# Text mining in long-term care: Exploring the usefulness of artificial intelligence in a nursing home setting

**Coen Hacking** [1,2]*, **Hilde Verbeek** [1,2], **Jan P. H. Hamers** [1,2], **Katya Sion** [1,2], **Sil Aarts** [1,2]

1 Faculty of Health Medicine and Life Sciences, Department of Health Services Research, CAPHRI Care and Public Health Research Institute, Maastricht University, Maastricht, The Netherlands, 2 The Living Lab in Ageing & Long-Term Care, Maastricht, The Netherlands

* c.hacking@maastrichtuniversity.nl

## Abstract

### Objectives

In nursing homes, narrative data are collected to evaluate quality of care as perceived by residents or their family members. This results in a large amount of textual data. However, as the volume of data increases, it becomes beyond the capability of humans to analyze it. This study aims to explore the usefulness of text mining approaches regarding narrative data gathered in a nursing home setting.

### Design

Exploratory study showing a variety of text mining approaches.

### Setting and participants

Data has been collected as part of the project *'Connecting Conversations'*: assessing experienced quality of care by conducting individual interviews with residents of nursing homes (n = 39), family members (n = 37) and care professionals (n = 49).

### Methods

Several pre-processing steps were applied. A variety of text mining analyses were conducted: individual word frequencies, bigram frequencies, a correlation analysis and a sentiment analysis. A survey was conducted to establish a sentiment analysis model tailored to text collected in long-term care for older adults.

### Results

Residents, family members and care professionals uttered respectively 285, 362 and 549 words per interview. Word frequency analysis showed that words that occurred most frequently in the interviews are often positive. Despite some differences in word usage, correlation analysis displayed that similar words are used by all three groups to describe quality of care. Most interviews displayed a neutral sentiment. Care professionals expressed a more

**Data Availability Statement:** The code, data and models discussed in the article will be made available at https://zenodo.org/record/6675839. Our interview data will not be publicly available due to the privacy of our participants. While certain

privacy-related information was removed from the transcripts (e.g. names of participants, living addresses, room numbers and other personally identifiable information), the stories that our participants tell are often of a personal nature. Upon request, our interview data can be provided with restrictions. For researchers who meet the criteria for access to confidential data. the data are available from the Living Lab in Ageing and Long-Term Care. Website: https://www.awolimburg.nl/, Email: ouderenzorg@maastrichtuniversity.nl, Phone: +31 (0)43 38 81570. Visiting Address: Duboisdomein 30 6229 GT Maastricht Postal Address: Academische Werkplaats Ouderenzorg Limburg Maastricht University Vakgroep Health Services Research - DUB 30 Postbus 616 200 MD Maastricht.

**Funding:** The author(s) received no specific funding for this work.

**Competing interests:** The authors have declared that no competing interests exist.

diverse sentiment compared to residents and family members. A topic clustering analysis showed a total of 12 topics including *'relations'* and *'care environment'*.

## Conclusions and implications

This study demonstrates the usefulness of text mining to extend our knowledge regarding quality of care in a nursing home setting. With the rise of textual (narrative) data, text mining can lead to valuable new insights for long-term care for older adults.

## Introduction

Patient perspectives have assumed a central role in various healthcare settings in the assessment of the quality of care [1,2]. For example, in nursing homes, the perspectives of residents, their family members, and care professionals are seen as an important prerequisite for the improvement of quality of care [3]. To obtain more in-depth information regarding the quality of care so that its essential features can be further explored, narrative data are collected. These data often contain not only the experiences of residents, but also information regarding their engagement, satisfaction, and quality of life [2,4]. Narrative data can be defined as any data that consist of stories concerning the lives of individuals (e.g., of residents) [5]. Examples of narrative data are interviews regarding the experienced quality of care or stories about the experience of older adults during the COVID-19 pandemic [6,7]. In addition, certain websites (e.g., Zorgkaart Nederland) allow long-term care receivers to post opinions on the quality of their care and their lives.

To date, several narrative data methods have been developed to evaluate the quality of care and the quality of life of residents [8–10]. In a research context, these often involve audio recordings and verbatim transcriptions thereof. Hence, these methods result in large amounts of unstructured textual data that cannot be analyzed manually. Ordinarily, they are analyzed using coding. This involves summarizing participants' quotes in several words that capture their essence [11,12]. It is a very time-consuming and tedious task and requires researchers to take a consistently objective approach. Consequently, large amounts of textual data regarding the quality of care gathered daily in nursing homes are only analyzed on a limited scale and with a limited scope. Since the amount of data using narrative methods is only expected to rise because of its expanding importance in science and health care in general, innovative analytical approaches are required.

Computers can process large quantities of data and deliver highly consistent results. Computerized methods, such as those found in data science, could offer a possible solution to overcome the difficulties of manual coding. Data science is a field aimed at extracting knowledge and insights from all kinds of data [13]. Text mining is a sub-category of data science that is directed at the retrieval, extraction, and synthesis of information from text [14]. Text mining includes methods such as frequency distributions, clustering, sentiment analysis, and visualization [14,15]. In contrast with manual analysis, text mining involves processes and methods that analyze textual data automatically. As a result, it is often used to go beyond the scope of specific projects. It is not intended to replace manual coding analyses, but it does provide a novel way to analyze large amounts of text.

Text mining is already being used to gain insights into the quality of care. For example, it is used in hospitals to process health care claims; to group medical records by patients' symptoms [16,17] or to predict the number of hospital admissions at an emergency department to avoid

overcrowding [18]. A more recent study showed an automated method for extracting clinical entities, such as treatment, tests, drugs and genes, from clinical notes [19]. Another study discussed how text mining can be used to assess the sentiment in tweets towards the COVID 19 pandemic [20]. These types of applications suggest that text mining could also be useful in processing narrative data in long-term care for older adults to acquire novel insights into the quality of care and quality of life. Therefore, the present study aimed to explore the usefulness of text mining approaches to textual data gathered in nursing home care.

## Methods

Several text mining methods were applied to examine narrative data collected within a nursing home setting. These narrative data consisted of interviews with residents of nursing homes, their family members, and their professional caregivers (i.e., triads). To enable automatic processing, all interviews were transcribed verbatim. Several preprocessing steps were applied. A variety of text mining analyses were conducted, including analysis of individual word frequencies and bigram frequencies, correlation analysis, and sentiment analysis. These are methods commonly used to gain insights into textual data [21].

### Data collection

The data were collected as part of the project *'Connecting Conversations,'* which assesses the quality of care by conducting separate interviews with residents of nursing homes, family members, and professional caregivers (i.e., triads) [3,4]. The underlying principles of *'Connecting Conversations'* are *'appreciative inquiry'* and 'relationship-centeredness' [3]. Appreciative inquiry means that a positive approach is used to focus on what is already good and helpful and to do it more frequently. Relationship-centered care means that the impact of relationships is integral to the process and outcomes of the care experience [22]. It has been shown that appreciative inquiry does not necessarily lead to a more positive conversation [23]. A total of n = 125 interviews were conducted at 5 different care organizations in the south of the Netherlands [8]. "The medical ethical committee of Zuyderland (the Netherland) approved the study protocol (17-N-86)." Information about the study was provided to all interviewers, residents, family members and caregivers by an information letter. All participants provided written informed consent: residents with legal representatives gave informed consent themselves (as well as their legal representatives) before and during the conversations.

A survey was conducted to establish a sentiment analysis model tailored to long-term care. Respondents were presented with a list of sentences that were randomly selected from the transcripts of *'Connecting Conversations.'* These sentences had to be assigned according to their sentiment: positive, neutral, or negative. The survey was anonymous, though participants were asked to provide some demographic details (i.e., age, gender, place of residence, and level of education). Participants were allowed to fill in the questionnaire at different times, and each time they were presented with different sentences.

### Data preprocessing

Preprocessing is a critical step in text mining. It involves the removal of any noisy and unreliable data from the original dataset [15]. Noise is erroneous information that makes the data more difficult to interpret [24]. An example of noisy data could be stuttering during a phone call or misspelled words in transcribed interviews. A lack of preprocessing could yield erroneous results. The product of data preprocessing is the final data set, which can then be analyzed for meaningful information. Preprocessing and analysis were performed using R and Python. R is a free software package for statistical computing and graphics [25], while Python is a free

software package for general-purpose computing [26]. The code for all the analyses can be found at https://zenodo.org/record/6675839.

To perform text mining analysis such as frequency distributions, correlation analysis, and sentiment analysis, the data were preprocessed in several steps: 1) all transcribed interviews were exported from Word files to Excel files, which were then loaded into R; 2) Words uttered by the interviewer were excluded from all the transcripts so that the results would not include them; 3) since many common words have no special significance (e.g., the, a, and is), removal of these so-called stop words was conducted by using a predefined list consisting of the most common Dutch stop words (n = 100); 4) since words in Dutch containing two letters are often stop words and/or non-informative, the minimum word length (i.e., the minimum number of characters that constitutes a word) was set to three; 5) a stemming approach was conducted to increase statistical significance for the various text mining methods. This approach groups words that refer to the same concept (e.g., nurses -> nurse, tummy -> stomach). It is similar to word embedding. Stemming is applied by computationally identifying the plurals and diminutives and reducing them to their root [27,28]. This rule-based approach is more conservative than word embedding approaches that group more generally related words (e.g., mother -> family, nurse -> care) [29,30].

## Data analysis

**Word frequencies.** To gain an initial understanding of the text, a frequency plot was conducted to visualize the individual words used most frequently in all the interviews (i.e., unigrams). Frequency plots can often be described by Zipf's law, which states that for any piece of text based on natural language, the frequency of any word is inversely proportional to its rank in the frequency distribution [15]. The (n = 50) words used with the highest frequency in the interviews are displayed in the frequency plots.

Bigram plots were also conducted. Bigrams [15] are combinations of two consecutive words such as *'very good.'* The sentence *'I love it here'* could therefore be split up into the bigrams *'I love,' 'love it,'* and *'it here.'* Bigrams can play a crucial role in text classification since they can capture the meanings of words that are not present when analyzing unigrams. For example, the word *'good'* has a different meaning when preceded by the word *'not.'* The 50 most frequent bigrams are displayed in terms of their relative frequency for all interviews with residents, family members of residents, and care professionals, respectively.

**Correlation analysis.** It is important to understand which words co-occur in the interviews with residents and family members, residents and care professionals, and care professionals and family members, respectively. Correlation analyses were therefore conducted to assess the correlation (i.e., co-occurrence) of words across the three groups. The words are displayed as a collection of data points, divided over three scatter plots. A log transformation was used for the x and y-axes to account for the skewness of the data (i.e., only some words occurred very frequently) [31]. Three bilateral Pearson correlation coefficients (r) were assessed for residents and family members, residents and care professionals, and care professionals and family members, respectively.

## Sentiment analysis

Sentiment analysis is the process of computationally identifying sentiment expressed in a piece of text [14,15,32]. For example, the sentence *'It's a good day'* could be identified as being positive, while the sentence *'It's a bad day'* could be identified as being negative: the sentence *'Today I went for a walk'* could be neutral, as it does not convey whether the walk is experienced as a positive or negative event.

Previous sentiment analysis models have been based on the general Dutch language [33]. However, as certain words can have a different meaning in a nursing home setting, the sentiment behind these words can also be different. For example, there is a negative sentiment behind the word *'plassen'* (*'peeing'*), whereas in the general Dutch language this world would be considered neutral. Therefore, a general sentiment model may not be suitable for analysis in long-term care.

Using the results of the survey, the sentiment for all occurring unigrams (single words such as *'good'* or *'very'*) and bigrams was calculated as a value between (-1) and (1). A value of (-1) denoted the most negative sentiment, a value of (1) denoted the most positive sentiment, and a value of (0) denoted a neutral sentiment. The calculation was based on how frequently a word was used in a positive or negative context. For example, a word that occurred 9 times as a positive word and once as a negative word was given a sentiment of 0.8 ((9 * 1 + 1 * -1) / 10 = 0.8). Since any sentiment larger than 0 could be defined as positive, a sentiment of 0.8 was positive. This analysis was carried out for all groups separately: residents (n = 39), family members (n = 37), and care professionals (n = 49), to show the difference between the three groups. The words that were used most often by each group were plotted with the corresponding sentiment of each word.

To show a broad overview of the differences in the overall experienced quality of care between the different groups, sentiment analysis on each interview was conducted [29]. This was illustrated by plotting the proportion of positive, negative, and neutral sentiments for each interview using a ternary plot [34]. A ternary plot is a triangular plot capable of displaying three variables that sum up to the same value [30]. In the case of the present sentiment analysis, the percentages of positive, negative, and neutral sentiments in every interview summed up to 100%.

## Topic clustering analysis

Topic clustering is the process of identifying topics (i.e. themes) in text segments and clustering (i.e. grouping) them together based on those topics [35,36]. Clustering is similar to a qualitative coding process using a grounded approach, as topics are discovered without any prior knowledge [12]. To discover topic clusters within interviews, several steps were conducted. Firstly, for each utterance in the interviews, keywords were extracted using part-of-speech (POS) tagging [37,38]. POS tagging can identify whether a word is a noun (e.g. nurse, room), a proper noun (e.g. Sarah, Maastricht), a verb (e.g. walking, knitting), or any other type. By extracting nouns as keywords, it is possible to discover overarching semantic topics. Secondly, word2vec was used to calculate similarities in words [29,30]. Word2vec is an algorithm that creates a vector (i.e. a point in high-dimensional space) for each word in such a way that words that are similar are located together [29]. Lastly, k-means was used to create k clusters of keywords [35]. For example, with k = 2, two clusters could be discovered that correspond to topics 'food' and 'family'. A value for k was calculated using the elbow method [36]. The elbow method balances between having many clusters which are too specific and having few clusters which are too general. For each cluster a topic name was manually assigned, based on the keywords belonging to that specific cluster. For example, if a cluster contained keywords such as 'mother', 'father' and 'daughter', the topic description was formulated as 'family'.

## Results

In total, 125 interviews were analyzed: 39 with residents, 37 with family members, and 49 with staff. A total of n = 202,986 words were uttered. Residents uttered 284.9 words per interview, family members 362.1 words, and care professionals 548.7 words.

## Word frequencies

Fig 1 shows the distribution of the most frequently used words in the interviews. A typical Zipf's law pattern is visible, indicating that the most commonly used words accounted for almost half of all the words in the interviews [15]. The frequency of any word was inversely proportional to its rank in the frequency distribution (i.e., the rank-frequency distribution was an inverse relation). The word *'goed'* (*'good'*) was among the most frequently used words,

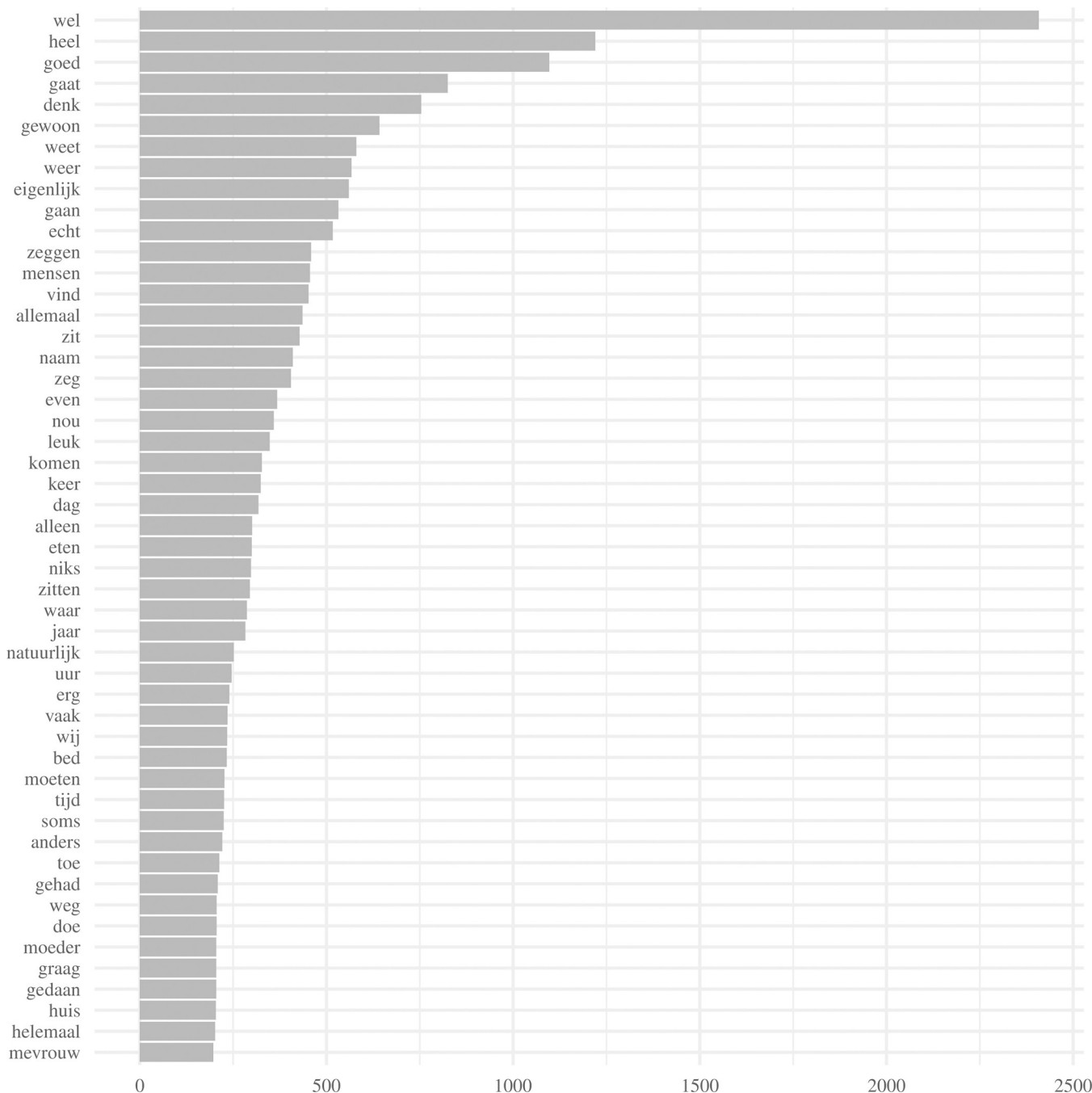

**Fig 1. First (n = 50) most frequently occurring unigrams across all interviews.**

indicating that the interviewees referred to many positive aspects. Moreover, words such as *'eten'* (*'food'*) and *'moeder'* (*'mother'*), provided insights into topics that participants perceived as important aspects of quality care. The word *'food'* was also mentioned frequently, suggesting that eating was another important topic. The frequent use of the word *'mother'* may refer to such residents being mentioned by family members. For example, residents' family members often referred to residents as their *'mother,'* indicating that the individual concerned was often a woman.

Since bigrams often contain more contextual information than unigrams, a bigram plot was conducted (Fig 2). The most frequently used bigrams included *'heel erg'* (*'very'*), *'heel goed'*

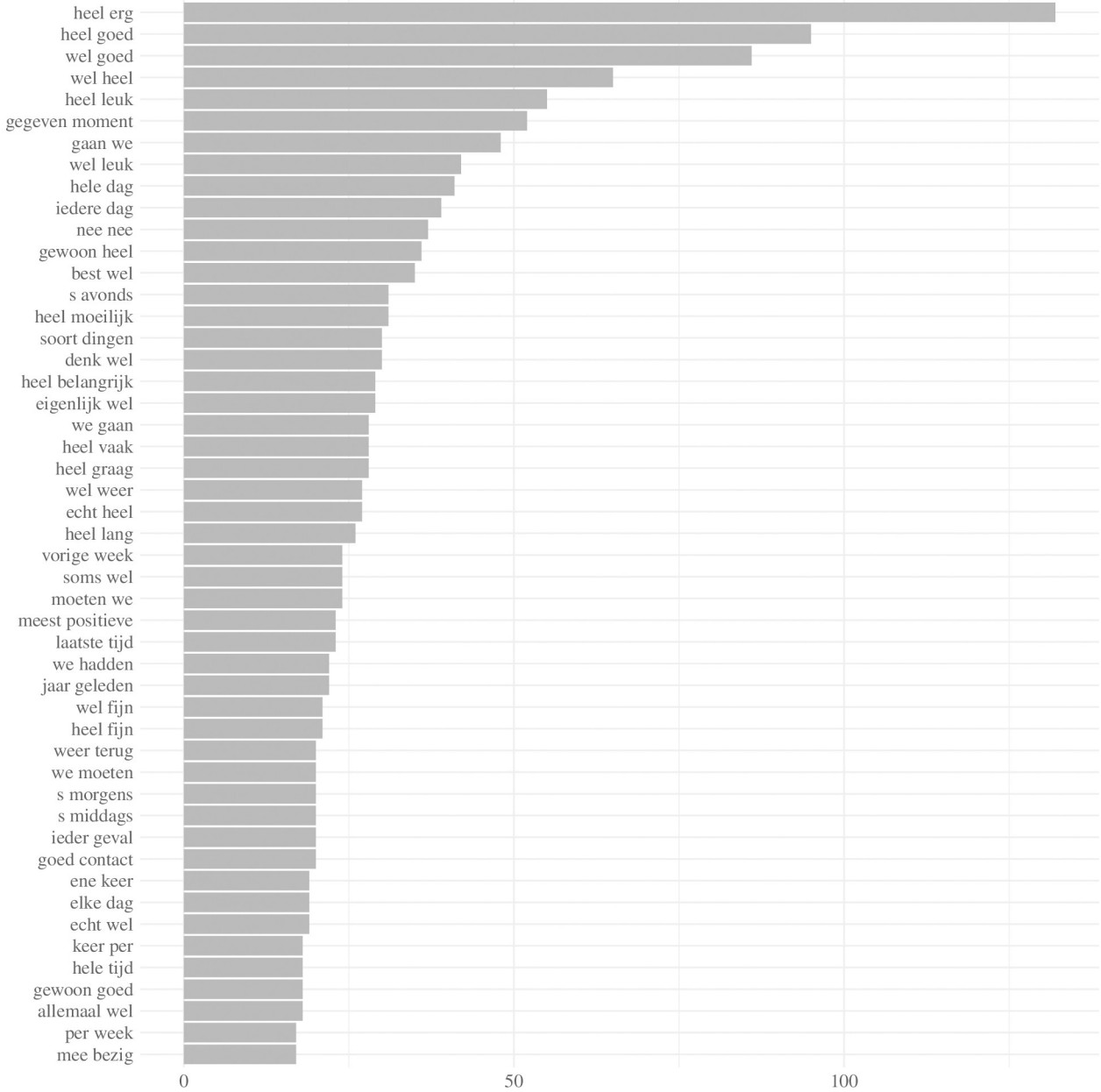

**Fig 2. First (n = 50) most frequently occurring bigrams across all interviews.**

(*'very well'*), *'wel goed'* (*'good'*), *'wel heel'* (*'very well'*), and *'wel leuk'* (*'nice'*). The bigram *'heel erg'* (very) was the most common bigram across all groups. This was used both negatively (e.g. *'very bad'*) and positively (e.g., *'very good'*). A sensitivity analysis of the words that proceeded this particular bigram revealed that the top 3 were *'goed,'* (*'good,'* n = 9), *'leuk'* (*'nice,'* n = 7) and *'tevreden'* (*'satisfied,'* n = 7). By contrast, the word *'slechts'* (*'bad'*) only proceeded the bigram *'heel erg'* (*'very'*) once.

## Correlation

Fig 3 shows the correlation plots between word usage among the three groups, where each point represents a word. For some points, the English translation of the corresponding word is also shown. All three plots are similar. First, they each display an uphill pattern which is indicative of a positive relationship between the two variables. Second, the majority of words form a pattern around the red diagonal line, resulting in a high correlation coefficient – r = 0.91, r = 0.83, and r = 0.92, respectively (i.e., word usage was largely similar across all groups). The further away the points are from the red diagonal line, the more different the word usage between groups and perhaps each group's perception of the quality of care. For example, words such as *'hobby'* (*'hobby'*), and *'dochter'* (*'daughter'*) occurred more frequently in the interviews with the residents than with family members. This suggests that these subjects were more important to the residents, either negatively or positively. Family members used words such as *'instelling'* (*'institution'*) and *'zaterdag'* (*'Saturday'*) more frequently.

## Sentiment analysis

A total of 234 participants assigned a sentiment to 11,519 sentences, which was 56% of the total. The mean age of all participants was 41 (SD: 13.7). Of all participants, 71% were women. Sixty-seven percent of the participants had at least a master's *degree, while 21% had a bachelor'*s degree; 67% said that they were currently living in the south of the Netherlands.

A scatter plot of the top 40 most commonly occurring words for residents, care professionals, and family members is displayed (Fig 4). The x-axis shows the sentiment value for each word between -1 (most negative) and 1 (most positive): the y-axis displays the frequency with which these words occurred. Many of the most common words were similar between residents, their family members, and care professionals. The words *'wel'* (*'well'* or *'quite'*), *'heel'* (*'very'*) and *'goed'* (*'good'*) occurred with very high frequency across all groups, but the sentiment was weakly positive. Words such as *'leuk'* (*'nice'*) and *'fijn'* (*'nice'*) occurred with high frequency with a strong positive sentiment; *'muziek'* (*'music'*) and *'activiteiten'* (*'activities'*) occurred with high frequency with a weakly positive sentiment.

## Sentiment analysis in triads

To illustrate the expressed sentiments of residents, family members, and care professionals, a ternary plot was created (Fig 5). It is 3-dimensional, with a positive, negative, and neutral axis. Each point in the triangle represents either a resident (red), a family member (blue), or a care professional (green). As can be seen, most conversations are closely grouped together and slightly above the absolute middle, meaning that they are mostly neutral and almost equally positive and negative. Residents represent the densest group, implying that they expressed rather similar sentiments, that is, neutral, with equal amounts of positive and negative sentences. Family members have a lower density, implying that they expressed a slightly more diverse range of sentiments; this group was on average more negative regarding the quality of care than the other groups. Care professionals cover the largest area and thereby were the most

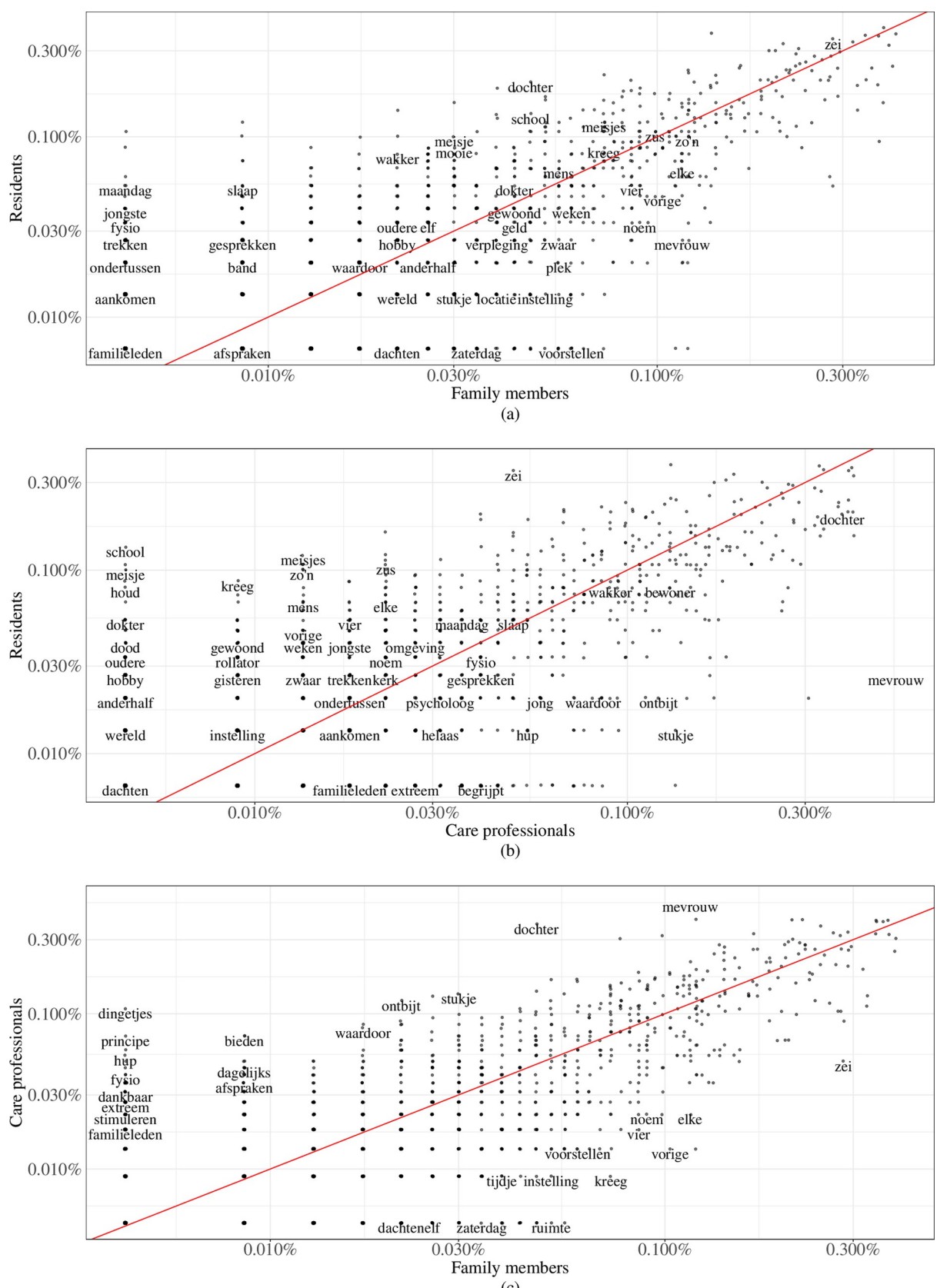

**Fig 3.** Bilateral correlations between the words of the 3 groups, respectively: (a) residents and family members; (b) residents and care professionals; and (c) care professionals and family members.

diverse in terms of sentiment expressed; this group was on average more positive regarding the quality of care than the other groups.

### Topic clustering analysis

To the topic clustering analysis is displayed in Fig 6; each keyword is represented as a dot. The axes have no real-world meaning; they are artificially created to highlight the difference between clusters (i.e. topics) [39]. Only the distance between dots has meaning: dots, and thus keywords, which are closer together are semantically more similar compared to dots that are further apart [29,39]. Dots with the same color belong to the same cluster. Although most key-words belong to one overarching cluster 'quality of care; (which was expected, as all keywords are related to experienced quality of care in a nursing home setting), utterances still show nuanced differences, leading to the discovery of 12 different, but related topics.

For each different cluster the topic name, the most important keywords and the number of sentences that belong to that topic are displayed (see Table 1). Certain clusters are well-defined such as *'health'* and *'food'*, while others display more overlap, such as *'care environment'*. This corresponds with Fig 6, as it shows certain clusters overlap very little with other clusters. The clusters *'relations'*, *'time'* and *'life experiences'* are the clusters with the most occurring keywords.

### Discussion

The present study aimed to explore the usefulness of text mining approaches regarding narra-tive data gathered in a nursing home setting. The textual information that was automatically gathered from the 125 interviews generated novel insights into the quality of care.

The results showed that the word *'good'* was among the most frequently used words in the interviews, which could indicate that, in general, participants had a positive experience of care. However, it might be argued that individual words have different meanings when preceded by different words (e.g., the word *'good'* can be preceded by the word *'not'* or *'very,'* thereby giving it a different sense). Hence, bigrams (i.e., groups of two consecutive words) were analyzed, and this revealed that the word '*good'* was often preceded by adjectives, indicating magnitude (e.g., *'very good'* or *'very nice'*). These word combinations frequently occurred in the interviews in all three groups, indicating positivity towards the quality of care. Previous research has demon-strated that, when conducting a manual sentiment analysis, words such as *'good'* are indicative of a positive experience regarding quality of care [8]. Correlation analysis showed that the same set of words were used by residents, their family members, and care professionals when discussing the quality of care. These findings imply that the three groups talked about similar topics when discussing the substantive issue. Sentiment analysis highlighted several positive words, including *'muziek'* (*'music'*) and *'activiteiten'* (*'activities'*). Because these words were used frequently in the interviews, it may be inferred that *'music'* and *'activities'* were regarded as important criteria for judging the quality of care [40]. In addition, sentiment analysis showed that the majority of interviews expressed mainly neutral sentiments, though the care professionals were more diverse and positive in their sentiments compared with the other groups [8]. This finding is underscored by previous research that has shown that, in general, care professionals are often more positive than residents or residents' family members con-cerning the quality of care the residents receive [8,41]. A topic clustering analysis yielded a

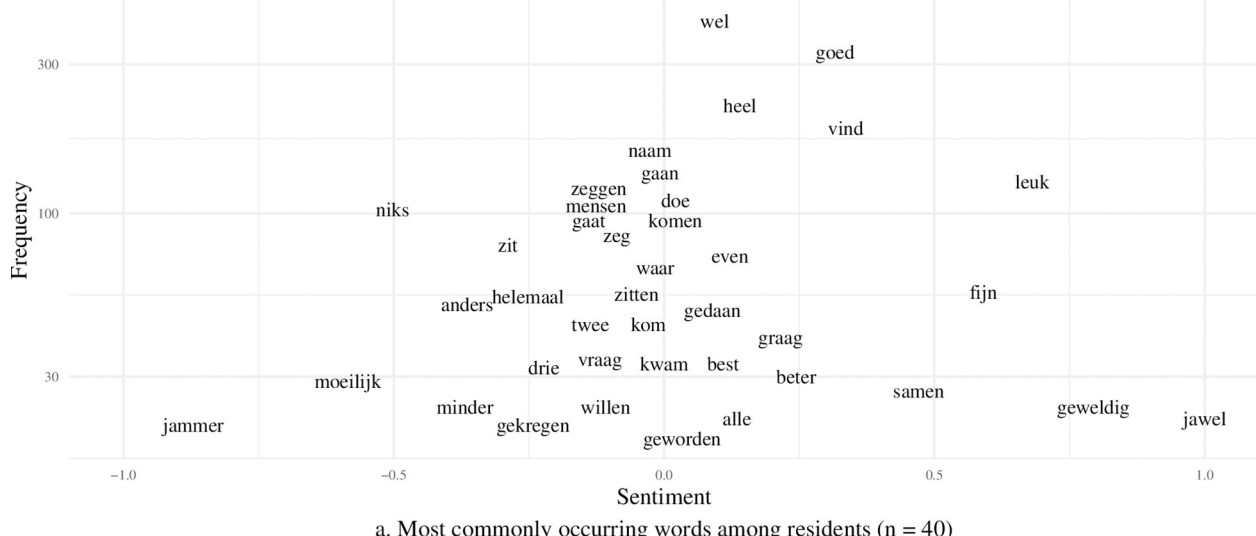

a. Most commonly occurring words among residents (n = 40)

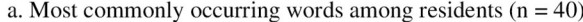

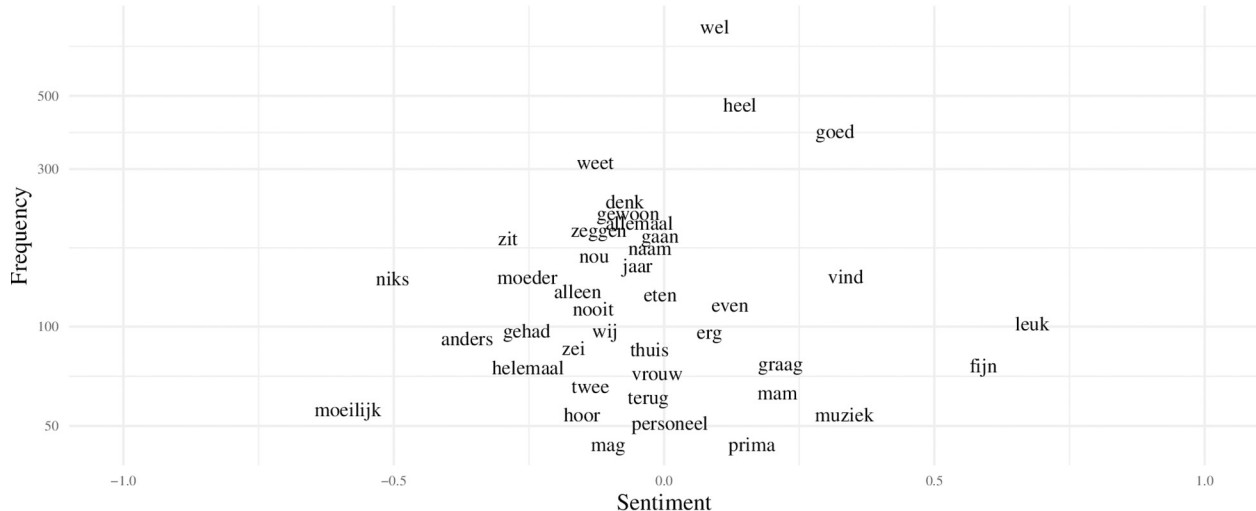

b. Most commonly occurring words among family members (n = 40)

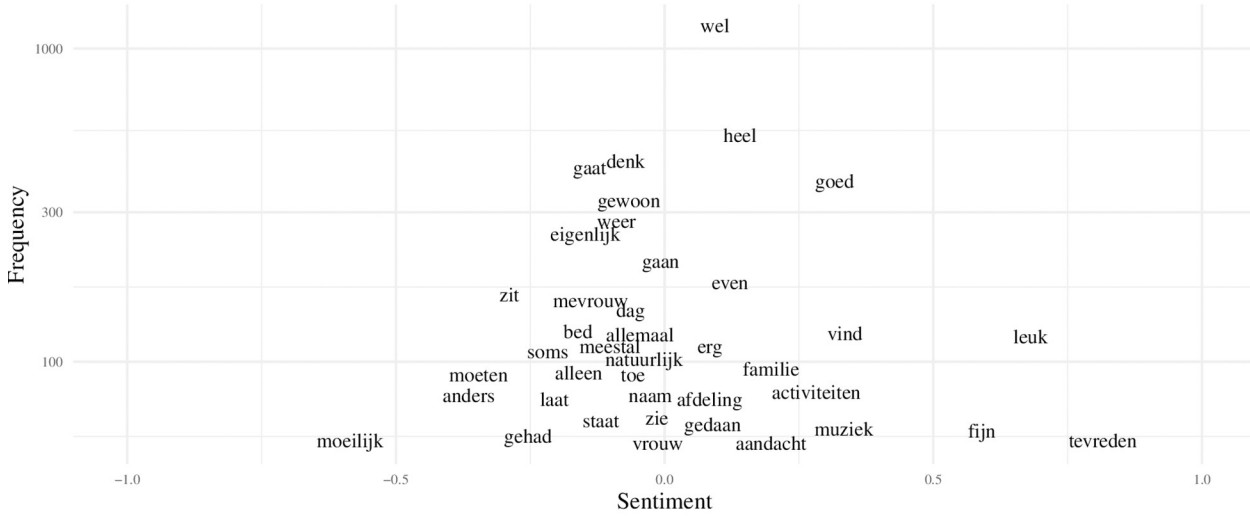

c. Most commonly occurring words among care professionals (n = 40)

**Fig 4.** Sentiment analyses: Scatter plots of positive and negative words used most often by (a) residents; (b) family members; and (c) care professionals.

variety of topics: while some topics were very clearly defined, including topics such as *'food'* and *'health'*, others were less clearly defined (e.g. *'miscellaneous'*). The large number of occurring keywords related to the topics *'relations'*, *'life experiences'* and *'care environment'* not only highlights the importance of these two topics in relation to experienced quality of care within a nursing home setting [3,8,42,43], but also underscores the validity of the text-mining approach.

The present study is the first to assess quality of care in a long-term care context by analyzing qualitative data through text mining. Making use of the vast amount of text in this way has given a voice to residents, their family members, and care professionals working in nursing homes. However, the study also has several possible limitations. Firstly, the word and bigram frequency analyses only contain absolute numbers. This analyses still contains words and bigrams which have little significance, i.e. words such as *'well'* or *'quite'*, which are less informative. Another limitation is the explainability of the sentiment analysis model which is a

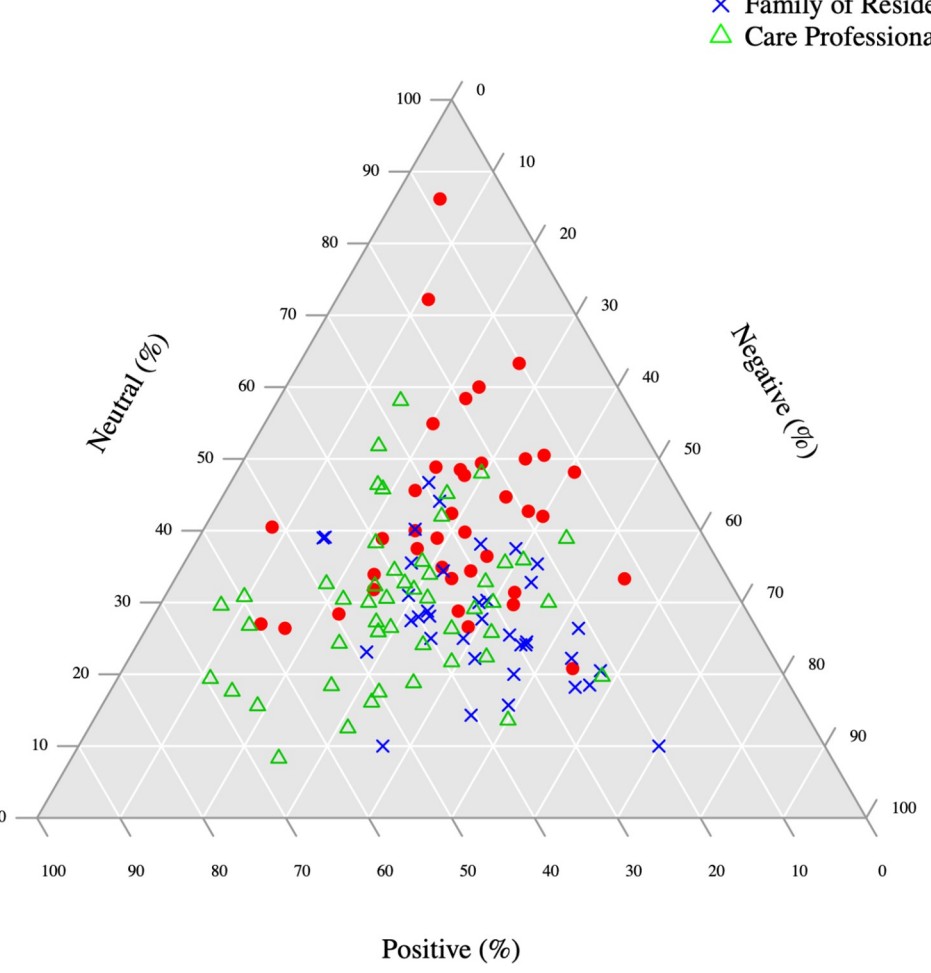

**Fig 5. An overview of the ratios (positive, negative and neutral) for all transcripts.**

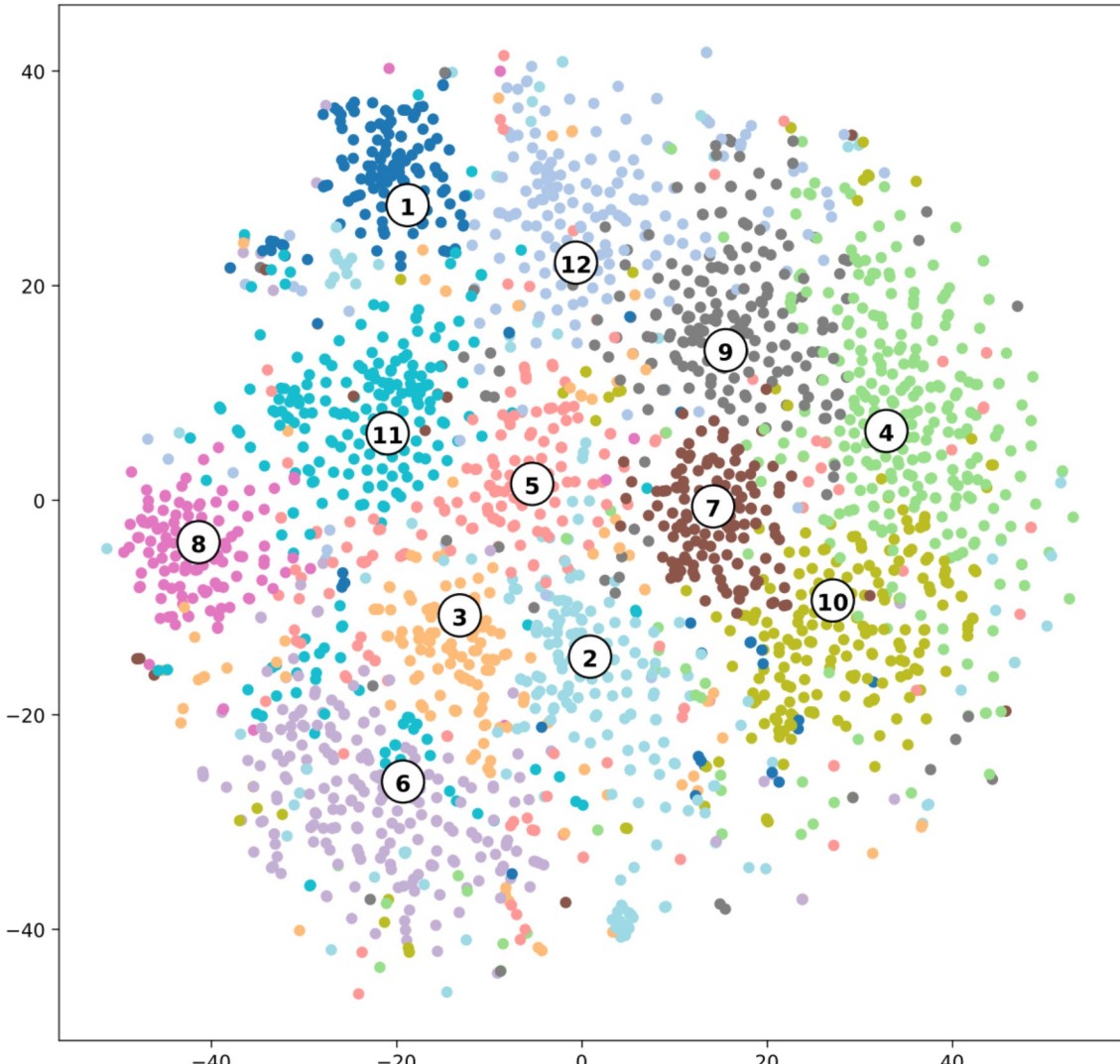

**Fig 6. A visualization of the topic clustering analysis.** Clusters are represented by numbers which correspond with the numbers in Table 1.

'deep learning' model. Deep learning is the optimization of large models for tasks such as sentiment analysis [32,37]. While sentiment analysis conducted using a deep learning method often results in more accurate results compared to machine learning models, these latter models can be explained more easily, as unigrams or n-grams (sequences of words) correspond with a certain sentiment prediction. With deep learning models, sentiment can be based on how every word relate to every other word in a text segment. These word relations are calculated from large text datasets and involves many abstract values [37,43].

## Future work

Future research could focus on combining narrative data with more quantitative measures related to, for example, the prevalence of care problems [44]. This could be achieved by using a text mining approach and various predictive algorithms. It would then be possible to relate narrative data regarding the quality of care to particular care issues (e.g. incontinence and malnutrition), thereby providing a more comprehensive view of quality of care.

**Table 1. Topics from the cluster analysis including the corresponding number of keywords occurring in the interviews (n = 125).**

| # | Topic name | Example keywords[a] | # Occurrences |
|---|---|---|---|
| 1 | Relations | Daughter, mother, family, children, man | 1995 |
| 2 | Activities | Flower arranging, television, Christmas, music, movie | 996 |
| 3 | Time | Time, start, hour, day, afternoon, week | 2175 |
| 4 | Care organization | Decision, matter, system, organizations, privacy | 1072 |
| 5 | Daily experiences | Meaning, moment, experience, progression, private | 525 |
| 6 | Physical nursing home environment | Room, neighborhood, door, ambulance, walker | 1106 |
| 7 | Health | Parkinson, hallucinations, miscarriage, eye drops, incident | 423 |
| 8 | Food | Dinner, desert, coffee, sandwich, potatoes | 527 |
| 9 | Life experiences | Life, moment, story, feeling, event | 1827 |
| 10 | Care environment | Nursing home, care, education, help, somatic | 1432 |
| 11 | Physical Appearance | Sewing machine, toe, hands, clothes, nightgown | 860 |
| 12 | Miscellaneous | A little, word, small error, things, remainder | 868 |

[a] Words were translated from Dutch into English. Only the English words are displayed.

Future research could also aim at further exploring the text-mining approaches used in the current study. By comparing the text mining approach against the current gold standard of manual coding, the text-mining approach could not only be validated but perhaps also improved. For example, by further improving topic clustering, it may become possible to automate the processes of qualitative coding (i.e. the analyses of qualitative data). As a consequence, analyzing qualitative data may become less time-consuming and more objective.

## Conclusion

To make use of the ever-growing amount of textual data related to the quality of care in long-term older persons' care, innovative and efficient methods are needed. The present study demonstrates the usefulness of a text mining approach to extend our knowledge regarding the quality of care in a nursing home setting. With the shift to more collections of textual (narrative) data, text mining in long-term elderly care can lead to valuable new insights that would not have been found using manual analysis.

## Acknowledgments

The authors would like to thank all the contributors of Python, R and the R and Python packages that were used for this study (a full list can be found at https://zenodo.org/record/6675839). Especially thanks to the contributors of the following packages: *'ggplot package'*, *'tm package'*, *'stopwords package'*, *'Ternary package'*, and the *R-markdown package*.

## Author Contributions

**Conceptualization:** Coen Hacking, Hilde Verbeek, Jan P. H. Hamers, Sil Aarts.

**Data curation:** Katya Sion.

**Formal analysis:** Coen Hacking, Sil Aarts.

**Investigation:** Coen Hacking, Katya Sion.

**Methodology:** Coen Hacking, Katya Sion, Sil Aarts.

**Project administration:** Sil Aarts.

**Resources:** Hilde Verbeek, Jan P. H. Hamers.

**Software:** Coen Hacking, Sil Aarts.

**Supervision:** Hilde Verbeek, Jan P. H. Hamers, Sil Aarts.

**Validation:** Sil Aarts.

**Visualization:** Coen Hacking, Sil Aarts.

**Writing – original draft:** Coen Hacking.

**Writing – review & editing:** Hilde Verbeek, Jan P. H. Hamers, Katya Sion, Sil Aarts.

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
