## [Decision Letter · Decision Letter 0]

7 Jan 2022

PONE-D-21-18516Text mining in long-term care: exploring the usefulness of computer-aided methods of analysisPLOS ONE

Dear Dr. Hacking,

Thank you for submitting your manuscript to PLOS ONE. After careful consideration, we feel that it has merit but does not fully meet PLOS ONE’s publication criteria as it currently stands. Therefore, we invite you to submit a revised version of the manuscript that addresses the points raised during the review process.

The reviewers raised a number of concerns with the manuscript that must be addressed. These include but are not limited to their view that the study should contain further empirical experiments and the fact that the study should be more clearly placed in the context of the existing body of work. The reviewers' comments can be viewed in full, below.

We look forward to receiving your revised manuscript.

Kind regards,

Natasha McDonald, PhD

Associate Editor

PLOS ONE

Journal Requirements:

2. Please provide additional details regarding participant consent. In the ethics statement in the Methods and online submission information, please ensure that you have specified (1) whether consent was informed and (2) what type you obtained (for instance, written or verbal, and if verbal, how it was documented and witnessed). If your study included minors, state whether you obtained consent from parents or guardians. If the need for consent was waived by the ethics committee, please include this information."

3. Thank you for including your ethics statement:  "Written ethical approval was provided (17-N-86 METC Zuyderland).".  

 For additional information about PLOS ONE ethical requirements for human subjects research, please refer to http://journals.plos.org/plosone/s/submission-guidelines#loc-human-subjects-research..

Reviewers' comments:

Reviewer's Responses to Questions

**Comments to the Author**

1. Is the manuscript technically sound, and do the data support the conclusions?

Reviewer #1: No

Reviewer #2: Yes

Reviewer #3: Yes

Reviewer #4: Yes

2. Has the statistical analysis been performed appropriately and rigorously? 

Reviewer #1: No

Reviewer #2: Yes

Reviewer #3: Yes

Reviewer #4: Yes

3. Have the authors made all data underlying the findings in their manuscript fully available?

Reviewer #1: No

Reviewer #2: Yes

Reviewer #3: No

Reviewer #4: Yes

4. Is the manuscript presented in an intelligible fashion and written in standard English?

Reviewer #1: No

Reviewer #2: Yes

Reviewer #3: Yes

Reviewer #4: Yes

5. Review Comments to the Author

Reviewer #1: This study certainly does not represents both scientific and practical interest after complete revision according to my point of view the paper is not suitable for publication in this journal so it must be rejected

Reviewer #2: Very interesting research. An innovative approach to the subject. I believe that the work should continue. They raise a very important topic. Residents, family members and care professionals uttered respectively 285, 362 and 549 words per interview. Word frequency analysis showed that words that occurred most frequently in the interviews are often positive.

Reviewer #3: I would like to thank the authors for their contribution. The study brings interesting insights, and it is also good to see that the authors are aware of possible limitations, which is certainly appreciated. Please consider my feedback below for the next version.

(1)

The study presents interesting exploratory analysis, indeed. However, I see that there is a good opportunity for applying further empirical experiments. For example, data clustering or developing a classification model should be a valuable addition. In this way, the study would be properly positioned within the context text or data mining in general.

(2)

Further, the study should also be positioned in line with the state-of-the-art methods of Natural Language Processing (NLP). The NLP research is currently dominated by the use of transformer models (e.g BERT). Further, there are specialized models (e.g. BioBERT), which might be quite applicable in the present work. In the recent years, earlier methods, such as bag-of-words, have been losing ground to transformers. Therefore, I believe that it is so important for the study to demonstrate that the authors are aware of such advances, which may be adopted as part of the future work, at least.

(3)

The introduction should refer to more recent work discussing or applying NLP in the healthcare context. For example:

https://doi.org/10.1109/JBHI.2021.3099755

http://dx.doi.org/10.5220/0010414508250832

(4)

Furthermore, I recommend touching on the explainability aspect. Model explainability is currently a very active area of research in the healthcare domain in particular. I suggest that the authors may discuss that aspect as part of the future work as well.

(5)

As a minor issue, I find the title a bit vague, the use of “computer-aided analysis” sounds too broad. I recommend using more focused terminology.

Reviewer #4: The objective of this research work is to evaluate the quality of care at a nursing home from the comments(narrative data) provided by the residents or their family members and care professionals as perceived by them. Text mining techniques were employed for analysis. Sentiment analysis was done to identify the emotions associated with these people from the words they uttered. The results of sentiment analysis could have been quantified as the authors rightly pointed out that it would be more significant to conclude the quality of care at the nursing home.

6. PLOS authors have the option to publish the peer review history of their article (what does this mean?). If published, this will include your full peer review and any attached files.

Reviewer #1: No

Reviewer #2: No

Reviewer #3: No

Reviewer #4: **Yes: **Dr.Sarojini Balakrishnan

---

## [Author Response · Author response to Decision Letter 0]

10 Feb 2022

Editor’s comments

We have incorporated the styling guidelines and adjusted the manuscript accordingly.

2. Please provide additional details regarding participant consent. In the ethics statement in the Methods and online submission information, please ensure that you have specified (1) whether consent was informed and (2) what type you obtained (for instance, written or verbal, and if verbal, how it was documented and witnessed). If your study included minors, state whether you obtained consent from parents or guardians. If the need for consent was waived by the ethics committee, please include this information."

We have revised these statements to include the necessary information: information about the study was provided to all interviewers, residents, family members and caregivers in advance by an information letter. All participants provided written informed consent: residents with legal representatives gave informed consent themselves (as well as their legal representatives) before and during the conversations.

3. Thank you for including your ethics statement: "Written ethical approval was provided (17-N-86 METC Zuyderland).". 

The statement has been amended. The medical ethics committee of Zuyderland, the Netherlands, approved the study protocol (17-N-86) and concluded that the study was not subject to the Medical Research Involving Human Subjects Act.

 For additional information about PLOS ONE ethical requirements for human subjects research, please refer to http://journals.plos.org/plosone/s/submission-guidelines#loc-human-subjects-research..

The statement has been amended here as well.

Comments to the Author

Have the authors made all data underlying the findings in their manuscript fully available?

Reviewer #1: No

Reviewer #2: Yes

Reviewer #3: No

Reviewer #4: Yes

We have addressed this comment by sharing our sentiment analysis model and the source data of our figures (i.e. unigram frequencies (per group), bigram frequencies and sentiment of unigrams). These will become available on Github after publication. Moreover, all R and Python code will be become available on Github. Our interview data will not be publicly available due to the privacy of our participants. While certain privacy-related information was removed from the transcripts (e.g. names of participants, living addresses, room numbers and other personally identifiable information), the stories that our participants tell are often of a personal nature. Upon request, our interview data can be provided. 

5. Review Comments to the Author

We would like to thank all 4 reviewers for their comments and suggestions for the paper. Three reviewers were very positive about our study and made concrete reflections and suggestions for improvement. We have responded to these in our rebuttal below. One reviewer was negative about the paper although there was no explanation why and suggestions for improvement were lacking. 

Reviewer #1: This study certainly does not represent both scientific and practical interest after complete revision according to my point of view the paper is not suitable for publication in this journal so it must be rejected

We would like to thank the reviewer for his/her time. We’re sorry that we couldn’t satisfy your scientific and practical interest and we regret that you were unable to provide us any concrete comments regarding this decision.

Reviewer #2: Very interesting research. An innovative approach to the subject. I believe that the work should continue. They raise a very important topic. Residents, family members and care professionals uttered respectively 285, 362 and 549 words per interview. Word frequency analysis showed that words that occurred most frequently in the interviews are often positive.

We would like to thank the reviewer for his/her positive words regarding our manuscript. We believe that the manuscript improved because of the changes made upon all reviewers’ comments. 

Reviewer #3: I would like to thank the authors for their contribution. The study brings interesting insights, and it is also good to see that the authors are aware of possible limitations, which is certainly appreciated. Please consider my feedback below for the next version.

(1) The study presents interesting exploratory analysis, indeed. However, I see that there is a good opportunity for applying further empirical experiments. For example, data clustering or developing a classification model should be a valuable addition. In this way, the study would be properly positioned within the context text or data mining in general.

The reviewer stresses and important point here: more analyses related to data clustering or classification would be a valuable addition. However, these are beyond the scope of this paper. The current manuscript has a different focus: to introduce the nursing/long-term care community, to the possibilities of AI, and more specifically, to text-mining. Hence, we believe that an additional analysis would make the article less comprehensible. Therefore, we believe this could better be addressed in a follow-up article. As a matter of fact, we are currently working on various deep learning analyses with BERT-based language models. In a new study, we’ll be discussing different classification analyses in detail and comparing them against a human baseline (e.g. the human form of analyzing qualitative data: open and axial coding). We are currently conducting a thematic analysis through multi-label classification, using labels (i.e. themes) that are related to quality of care. 

(2) Further, the study should also be positioned in line with the state-of-the-art methods of Natural Language Processing (NLP). The NLP research is currently dominated by the use of transformer models (e.g BERT). Further, there are specialized models (e.g. BioBERT), which might be quite applicable in the present work. In the recent years, earlier methods, such as bag-of-words, have been losing ground to transformers. Therefore, I believe that it is so important for the study to demonstrate that the authors are aware of such advances, which may be adopted as part of the future work, at least.

As the reviewer states, a bag of words approach, such as the naïve bayes approach that we used in our study does not reach the same accuracy as modern transformer models. We initially chose a naïve bayes approach, because such a model is ‘easy’ to explain and it’s much easier to replicate than a deep learning approach. Replication of a transformer model is more difficult due to inference and training requiring many computational resources. We wanted to show that even a less advanced method could lead to interesting insights. Additionally, deep learning approaches may not always outperform traditional methods [3]. In our case, by using naïve bayes we achieved an accuracy of around 70%, whereas with a BERT-based deep learning approach we achieved 75%. As this does increase the reliability of our research, we’ve replaced the naïve bayes sentiment analysis with RobBERT which was fine-tuned for sentiment analysis using the interview data [2]. Often such models are fine-tuned on review data (e.g. book or movie reviews), however the language in such reviews is very different from the language used in our interview data. Because of this change, we’ve also mentioned the programming language Python in the methods section, as many deep learning packages depend on this programming language.

(3) The introduction should refer to more recent work discussing or applying NLP in the healthcare context. For example:

https://doi.org/10.1109/JBHI.2021.3099755

http://dx.doi.org/10.5220/0010414508250832

The manuscript now contains the first article the reviewer proposed: “An Accurate Deep Learning Model for Clinical Entity Recognition From Clinical Notes.” Additionally, we’ve included another recent study about natural language processing in a healthcare context: a study conducted in 2020 that discusses how a model such as RoBERTa can be used to assess the sentiment in tweets regarding the COVID 19 pandemic [1]. 

(4) Furthermore, I recommend touching on the explainability aspect. Model explainability is currently a very active area of research in the healthcare domain in particular. I suggest that the authors may discuss that aspect as part of the future work as well.

We’ve dedicated an additional paragraph to model explainability as part of the future work. In this section we discuss the explainability of both traditional machine learning approaches and deep learning approaches. We also shortly discuss the advantages and disadvantages of both these approaches regarding the explainability (page 16, line 315-321). 

(5) As a minor issue, I find the title a bit vague, the use of “computer-aided analysis” sounds too broad. I recommend using more focused terminology.

We’ve decided to change the title to: ‘Text mining in long-term care: exploring the usefulness of artificial intelligence in a nursing home setting’. 

Reviewer #4: The objective of this research work is to evaluate the quality of care at a nursing home from the comments(narrative data) provided by the residents or their family members and care professionals as perceived by them. Text mining techniques were employed for analysis. Sentiment analysis was done to identify the emotions associated with these people from the words they uttered. The results of sentiment analysis could have been quantified as the authors rightly pointed out that it would be more significant to conclude the quality of care at the nursing home.

Thank you for your review. We’ve made a number of improvements based on all reviewers’ feedback. The results of the sentiment analysis were quantified and visualized in Figure 5 of the manuscript.

References:

1. Azeemi AH, Waheed A. COVID-19 Tweets Analysis through Transformer Language Models. arXiv preprint arXiv:2103.00199. 2021 Feb 27. Available from https://arxiv.org/pdf/2103.00199

2. Wang, A., Singh, A., Michael, J., Hill, F., Levy, O., & Bowman, S. R. (2018). GLUE: A multi-task benchmark and analysis platform for natural language understanding. arXiv preprint arXiv:1804.07461. https://arxiv.org/pdf/1804.07461.pdf

3. Shwartz-Ziv, R., & Armon, A. (2022). Tabular data: Deep learning is not all you need. Information Fusion, 81, 84-90.

---

## [Decision Letter · Decision Letter 1]

17 Mar 2022

PONE-D-21-18516R1

Text mining in long-term care: exploring the usefulness of artificial intelligence in a nursing home setting

PLOS ONE

Dear Dr. Hacking,

Thank you for submitting your manuscript to PLOS ONE. After careful consideration, we feel that it has merit but does not fully meet PLOS ONE’s publication criteria as it currently stands. Therefore, we invite you to submit a revised version of the manuscript that addresses the points raised during the review process.

We look forward to receiving your revised manuscript.

Kind regards,

Haoran Xie

Academic Editor

PLOS ONE

Journal Requirements:

Reviewers' comments:

Reviewer's Responses to Questions

**Comments to the Author**

1. If the authors have adequately addressed your comments raised in a previous round of review and you feel that this manuscript is now acceptable for publication, you may indicate that here to bypass the “Comments to the Author” section, enter your conflict of interest statement in the “Confidential to Editor” section, and submit your "Accept" recommendation.

Reviewer #2: All comments have been addressed

Reviewer #3: (No Response)

Reviewer #4: All comments have been addressed

2. Is the manuscript technically sound, and do the data support the conclusions?

Reviewer #2: Yes

Reviewer #3: (No Response)

Reviewer #4: Yes

3. Has the statistical analysis been performed appropriately and rigorously? 

Reviewer #2: Yes

Reviewer #3: (No Response)

Reviewer #4: Yes

4. Have the authors made all data underlying the findings in their manuscript fully available?

Reviewer #2: Yes

Reviewer #3: (No Response)

Reviewer #4: (No Response)

5. Is the manuscript presented in an intelligible fashion and written in standard English?

Reviewer #2: Yes

Reviewer #3: (No Response)

Reviewer #4: Yes

6. Review Comments to the Author

Reviewer #2: I believe that the work fully deserves to be published in PLOS One. Original, innovative topic. I have no comments on the research work. Congratulations to the authors.

Reviewer #3: I thank the authors very much for their response, and for the amendments done.

However, I am afraid that I have to keep my initial recommendation that the manuscript would need further experiments to be considered for a journal publication, especially for a quite high-impact journal like PLOS ONE.

I find the argument about ”introducing the nursing care community to the possibilities of AI” not largely convincing. Applications of Text-Mining have already spanned a plethora of publications in the healthcare domain.

Reviewer #4: (No Response)

7. PLOS authors have the option to publish the peer review history of their article (what does this mean?). If published, this will include your full peer review and any attached files.

Reviewer #2: **Yes: **Przemysław Karol Wolak

Reviewer #3: No

Reviewer #4: **Yes: **Dr.Sarojini Balakrishnan

---

## [Author Response · Author response to Decision Letter 1]

25 Apr 2022

Reviewer #2: I believe that the work fully deserves to be published in PLOS One. Original, innovative topic. I have no comments on the research work. Congratulations to the authors.

We would like to thank the reviewer for his/her positive words regarding the originality and innovativeness of our work and corresponding manuscript. 

Reviewer #3: I thank the authors very much for their response, and for the amendments done.

However, I am afraid that I have to keep my initial recommendation that the manuscript would need further experiments to be considered for a journal publication, especially for a quite high-impact journal like PLOS ONE. I find the argument about ”introducing the nursing care community to the possibilities of AI” not largely convincing. Applications of Text-Mining have already spanned a plethora of publications in the healthcare domain.

We would like to thank the reviewer for his/her response. The authors decided to include a topic clustering analysis with the aim of discovering topics that were discussed in the interviews with residents, family members and care professionals. Keywords were extracted using POS tagging with a Dutch RobBERTa-like model. Nouns were selected to be representative of the topics existing in the interviews. Using word2vec, keywords were encoded as vectors. These vectors were transformed from cosine to Euclidian space and clustered using k-means. A value for k was calculated using the elbow method. A value of k = 12 was found to be optimal. Afterwards, each cluster was manually given a topic name based on the keywords that were included in the respective cluster. Based on this added analysis, the methods, results, and discussion were amended.

---

## [Editor Report · Decision Letter 2]

27 Apr 2022

Text mining in long-term care: exploring the usefulness of artificial intelligence in a nursing home setting

PONE-D-21-18516R2

Dear Dr. Hacking,

We’re pleased to inform you that your manuscript has been judged scientifically suitable for publication and will be formally accepted for publication once it meets all outstanding technical requirements.

Kind regards,

Haoran Xie

Academic Editor

PLOS ONE
---

## [Editor Report · Acceptance letter]

28 Jul 2022

PONE-D-21-18516R2 

Text mining in long-term care: exploring the usefulness of artificial intelligence in a nursing home setting 

Dear Dr. Hacking:

I'm pleased to inform you that your manuscript has been deemed suitable for publication in PLOS ONE. Congratulations! Your manuscript is now with our production department. 

Kind regards, 

on behalf of

Professor Haoran Xie 

Academic Editor

PLOS ONE